# Equalizer Parameters' Adjustment Based on an Oversampled Channel Model for OFDM Modulation Systems

Marcin Kucharczyk [1,*] , Grzegorz Dziwoki [1] , Jacek Izydorczyk [1] , Wojciech Sułek [1] , Adam Dustor [1] , Wojciech Filipowski [1] , Weronika Izydorczyk [1] , Piotr Kłosowski [1] , Piotr Zawadzki [1] , Piotr Sowa [2] and Michał Rajzer [3]

1   Department of Telecommunications and Teleinformatics, Silesian University of Technology, ul. Akademicka 16, 44-100 Gliwice, Poland; grzegorz.dziwoki@polsl.pl (G.D.); jacek.izydorczyk@polsl.pl (J.I.); wojciech.sulek@polsl.pl (W.S.); adam.dustor@polsl.pl (A.D.); wojciech.filipowski@polsl.pl (W.F.); weronika.izydorczyk@polsl.pl (W.I.); piotr.klosowski@polsl.pl (P.K.); piotr.zawadzki@polsl.pl (P.Z.)
2   iSowa.io Piotr Sowa, ul. Kościuszki 36d/15, 32-020 Wieliczka, Poland; piotr@isowa.io
3   Faculty of Automatic Control, Electronics and Computer Science, Silesian University of Technology, ul. Akademicka 16, 44-100 Gliwice, Poland; mr306081@student.polsl.pl
*   Correspondence: marcin.kucharczyk@polsl.pl

**Abstract:** A physical model of a wireless transmission channel in the time domain usually consists of the main propagation path and only a few reflections. The reasonable assumptions made about the channel model can improve its parameters' estimation by a greedy OFDM (Orthogonal Frequency Division Multiplexing) equalizer. The equalizer works flawlessly if delays between propagation paths are in the sampling grid. Otherwise, the channel impulse response loses its compressible characteristic and the number of coefficients to find increases. It is possible to get back to the simple channel model by data oversampling. The paper describes how the above idea helps the OMP (Orthogonal Matching Pursuit) algorithm estimate channel coefficients. The authors analyze the oversampling algorithm on the one hand to assess the influence of filtering function and signal resolution on the quality of the channel impulse response reconstruction. On the other hand, the abilities of the OMP algorithm are analyzed to distinguish components of the oversampled signal. Based on these analyses, we proposed modifications to the compressible channel's impulse response reconstruction algorithm to minimize the number of transmission errors. A distinction was made between the filters used in the OMP search and channel reconstruction stages before calculating equalizer coefficients. Additionally, the results of the search stage were considered as elements within the groups.

**Keywords:** OFDM; compressed sensing; compressible channel; oversampling; channel estimation

## 1. Motivation

The problem of channel equalization in OFDM (Orthogonal Frequency Division Multiplexing) transmission systems has been extensively discussed. One of the most straightforward solutions involves the use of an independent one-tap equalizer for each OFDM subchannel, calculated using the zero-forcing (ZF) algorithm based on the received pilot frame [1,2]. This approach is straightforward and practical, particularly when combined with the estimation of the SNR (Signal-to-Noise Ratio) in the equalization algorithm MMSE (Minimum Mean Square Error) [3,4]. More advanced equalization techniques can be found in the literature [5–7], but the most popular methods rely on prior estimation of channel characteristics in the frequency or time domain [8–11].

The portable transmission system based on SDR (Software-Defined Radio), which constitutes a crucial aspect of the research discussed in this paper, incorporates the MMSE algorithm. The primary objective is to improve the outcome of the equalization without requiring a complete system redesign. To achieve this, a refinement of channel estimation is proposed, taking advantage of the findings obtained from analyses of transmission channel

measurements. These measurements were made under actual operating conditions of the transmission equipment. A diagram of the system for collecting data is presented in Figure 1. More than 78,000 measurements were collected in various transmission environments, as detailed in Table 1.

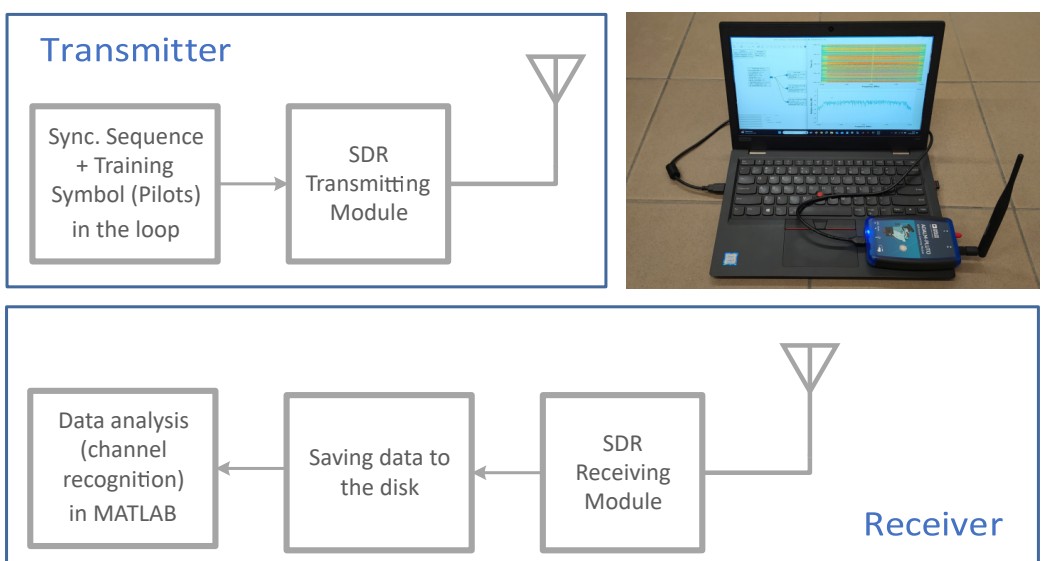

**Figure 1.** Diagram of system for collecting data for channel recognition (with a portable receiver in the photo).

**Table 1.** Information about database of measurements.

| Propagation Environment | Approx. Number of Measurements |
| --- | --- |
| Inside building | 25,000 |
| Between inside and outside of building | 15,000 |
| Outside building, in city environment | 38,000 |

Examination of the resulting database revealed that the primary problem lies in the presence of two closely spaced paths, leading to a significant null in the frequency response. Consequently, the research focused on using compressive sensing methodology [12,13] to explore the simple signal framework.

The impulse response of the two-path channel only has two coefficients that are not zero. In the case of band-limited systems, this is only valid if the delay difference between the paths is equal to the sampling period and if the transmitter and receiver are perfectly synchronized. When these conditions are met, iterative compressive sensing algorithms such as OMP (Orthogonal Matching Pursuit) [14,15] and CoSaMP (Compressive Sampling Matching Pursuit) [16] perform well. However, if the synchronization is not perfect, the observed channel response is the result of convolving the actual channel response (which has a limited number of coefficients) with the baseband filter response (which is theoretically infinite) of the transmission system [17,18].

The central concept proposed by the author is distinguishing between the actual channel response and the filtered response. This can be achieved by employing the compressive sensing algorithm to accurately recover the delay of the actual channel paths. To achieve this, the channel impulse response is oversampled in the equalizer [19,20], and a path search is conducted using the OMP algorithm on a signal with a resolution higher than the sampling frequency. The estimated impulse response of the channel is then filtered by the baseband filter, which models the appropriate response and provides coefficients for precise estimation of the equalizer.

This paper aims to enhance the channel recovery mechanism mentioned above to achieve a lower Bit Error Rate (BER) in the transmission system. We propose the following modifications to accomplish this.

1.  Instead of using the same filter parameters to search the path delay and calculate the channel impulse response, we use different filters. One filter is used to upsample the signal before the OMP algorithm, and another is used to downsample the signal back to the original sampling frequency.
2.  In the estimation stage of the OMP algorithm, we consider the delay values obtained in the search stage and the nearest neighbors. This is carried out by assuming that the actual path delay lies within that vicinity.

Furthermore, theoretical analyses are presented to justify the modifications introduced. The simulation results of the entire system validate the effectiveness of the proposed approach.

The following section provides a more comprehensive explanation of the problem of estimating the channel impulse response. Section 3 focuses on the compressive sensing algorithm employed for the oversampled signal. In the following section, we aim to estimate the error values resulting from the filters used in the transmission system and the calculations of equalizer coefficients. This leads to the formulation of the algorithm. Section 5 presents the simulation results of the modified transmission system with the proposed algorithm. Finally, the last section draws some conclusions.

The notation conventions for this problem formulation involve the combination of signals in both the time and frequency domains, as well as the incorporation of both discrete and continuous signals. To distinguish between signals in the frequency domain and those in the time domain, uppercase letters are used for the former and lowercase letters are used for the latter. In the case of discrete signals, the elements of the time domain are denoted by the index $n$, while the elements of the frequency domain are denoted by the index $k$. The letter $t$ typically represents continuous signals, although there are a few instances where it represents a discrete signal with extremely high resolution that is not explicitly defined.

## 2. Problem Statement

This research was based on a conventional OFDM transmission system, as shown in Figure 2. According to this scheme, the signal is transmitted and processed in the baseband. However, upon analyzing Figure 1, it can be concluded that the actual transmission takes place in the high-frequency band aligned with the configuration of the SDR module. This does not affect the presented processing model, although the shaping filter recognition problem presented later in the article also applies to this module.

The frame structure of the system included an additional OFDM symbol for channel equalization, placed after the synchronization sequence. This symbol consisted of a pilot in each subchannel, and the equalizer parameters were calculated entirely in the frequency domain. The OFDM system uses $K$ subchannels in the frequency domain for transmission, so the same number of pilots are present in the symbol $\mathbf{X}_{K \times 1} = [X_1, X_2, ..., X_k, ..., X_K]$ used for equalization. After applying the IFFT (Inverse Fast Fourier Transform), a cyclic prefix was added to the symbol $\mathbf{x}_{N \times 1}$ to create a signal vector in the time domain. This symbol was transmitted through a channel with a finite-duration impulse response $h(t)$ after being converted from digital to analog. The signal received includes additive white noise $n_0(t)$. In discrete time, the channel impulse response, shorter than the cyclic prefix, was represented by the vector $\mathbf{h}_{N \times 1}$ lengthened $N$. The receiver filters the signal, removes the cyclic prefix, and transforms the resulting $N$-element signal $\mathbf{y}_{N \times 1}$ into the frequency domain as $\mathbf{Y}_{K \times 1}$ using the FFT (Fast Fourier Transform). The transmitted and received symbols are divided in the frequency domain to estimate $K$ one-tap equalizers, which are stored in the vector $\mathbf{E}_{K \times 1}$ using the ZF algorithm [1]:

$$\mathbf{E}_{K \times 1} = \mathbf{X}_{K \times 1} \oslash \mathbf{Y}_{K \times 1} = \mathbf{1}_{K \times 1} \oslash \mathbf{H}_{K \times 1}. \tag{1}$$

Symbol $\oslash$ in the above formula denotes Hadamard division, so vectors are divided element by element:

$$E_k = \frac{X_k}{Y_k} = \frac{1}{H_k} \quad \text{for} \quad k = 1, 2, \ldots, K. \tag{2}$$

The ZF equalizer functions correctly when the actual channel impulse response length is shorter than the length of the cyclic prefix $P$ used by the system.

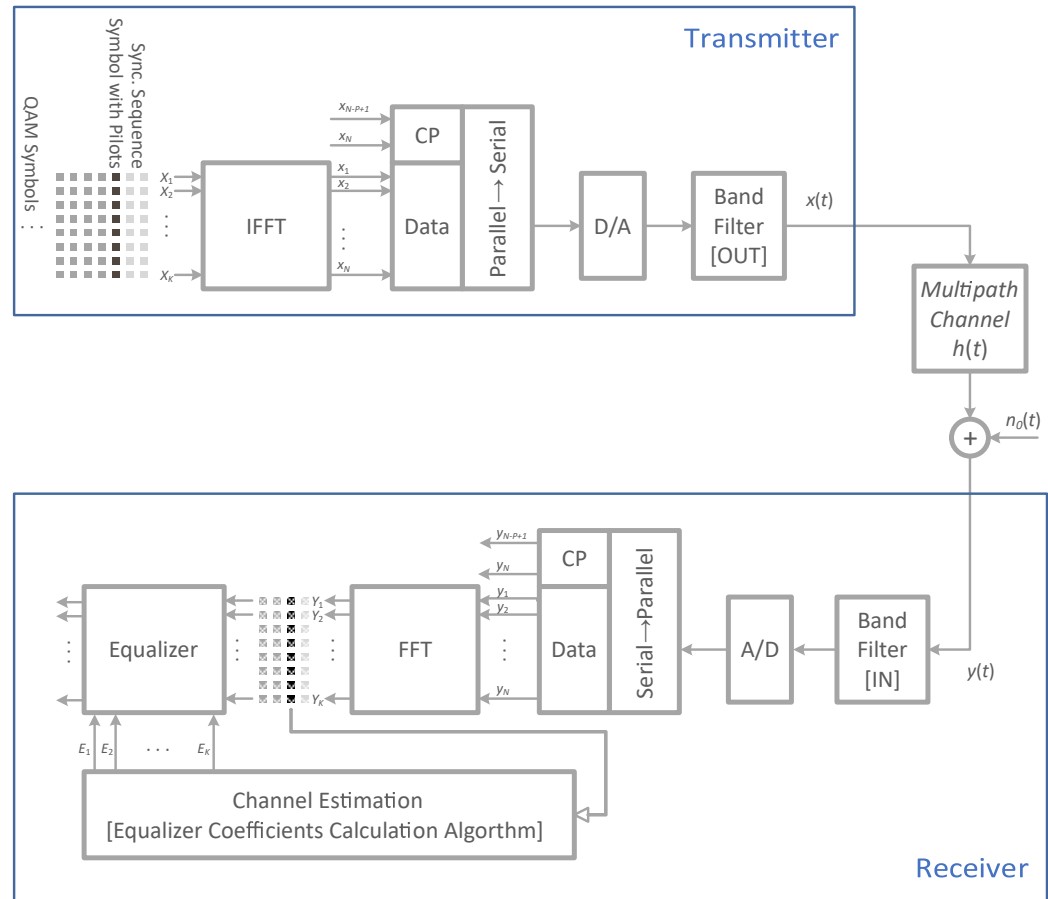

**Figure 2.** Diagram of the used transmission system with OFDM (Orthogonal Frequency Division Multiplexing) modulation.

Equation (1) does not rely on specific details about the physical characteristics of the channel. Previous studies [10,21,22] have shown that actual transmission channels can be accurately modeled as short FIR (Finite Impulse Response) filters, where the coefficients represent the different paths through which the signal propagates. Based on our measurements and information collected from the literature, it was determined that six paths adequately represent the channel.

As stated previously, upon closer examination of the channel data obtained from measurements of the transmission equipment utilized in our investigation, we observed that the signal exhibits a prominent path in most cases. However, a significant issue arises when the reflection from a nearby secondary path is received. This channel type, characterized by only two closely located paths, results in a deep zero in the channel's frequency response, rendering the subchannels useless for transmission. However, if the channel estimation results closely resemble its original shape, the probability of successful error correction in the FEC (Forward Error Correction) decoder [23–25] is enhanced.

The discrete impulse response of the channel in the frequency domain, denoted as $\mathbf{H}_{K \times 1}$ and obtained from (1), can be computed by applying the FFT to its time domain counterpart. Therefore, the training data defined by the pilots in the frequency domain

are sufficient to establish an equation that incorporates the time impulse response of the channel, represented by $\mathbf{h}_{N\times 1}$.

$$\mathbf{Y}_{K\times 1} = \mathbf{X}_{K\times 1} \odot \mathcal{F}(\mathbf{h}_{N\times 1}) = \mathbf{X}_{K\times 1} \odot \mathbf{F}_{K\times N}\mathbf{h}_{N\times 1} \tag{3}$$

or

$$\mathbf{H}_{K\times 1} = \mathcal{F}(\mathbf{h}_{N\times 1}) = \mathbf{F}_{K\times N}\mathbf{h}_{N\times 1}, \tag{4}$$

In this equation, the Hadamard product of vectors is denoted by $\odot$, and the FFT transform is represented by $\mathcal{F}$. The lengths of the vectors in the frequency domain ($K$) and in the time domain ($N$) are equal, typically determined by the size of the FFT used in the system, which corresponds to the size of the Fourier transformation matrix, denoted as $\mathbf{F}_{K\times N}$.

If the length of the channel impulse response in the time domain is less than $N$, the vector $\mathbf{h}_{N\times 1}$ will include elements that have a value of zero. Unknown elements of vector $\mathbf{H}_{K\times 1}$ in the frequency domain (channels without pilots) lead to the removal of elements of this vector and row in the matrix $\mathbf{F}_{K\times N}$.

The channel impulse response $\mathbf{h}_{N\times 1}$ is the unknown value in Equations (3) and (4). The resulting equalizer is a vector of length $N$, but as mentioned earlier, the channel impulse response is considerably shorter.

The formulas above do not include the presence of noise, but in reality, noise is present in the transmission system. Once the noise is considered, the equation that describes the system's operation is given by (5). In this equation, $\mathbf{e}_{N\times 1}$ and $\mathbf{E}_{K\times 1}$ represent the error values caused by the noise.

$$\mathbf{H}_{K\times 1} = \mathbf{F}_{K\times N}(\mathbf{h}_{N\times 1} + \mathbf{e}_{N\times 1}) = \mathbf{F}_{K\times N}\mathbf{h}_{N\times 1} + \mathbf{E}_{K\times 1}. \tag{5}$$

*2.1. FIR Filter Design*

Equation (4) represents an FIR filter $\mathbf{h}_{N\times 1}$ with a known frequency response $\mathbf{H}_{K\times 1}$. The matrix $\mathbf{F}_{K\times N}$ contains the base signals that represent the frequencies considered. Initially, the length of the FIR filter is limited to the size of the cyclic prefix $P$. The cyclic prefix is added to the signal to prevent intersymbol interference (ISI) and should have a length greater than the actual impulse response of the channel $h(t)$. Under the assumption of proper synchronization at the receiver, the non-zero coefficients of the channel impulse response are expected only in the first $P$ elements of the $\hat{\mathbf{h}}_{N\times 1}$ vector. Based on this assumption, it is possible to reduce the sizes of the matrices $\mathbf{F}_{K\times N}$ and $\hat{\mathbf{h}}_{N\times 1}$.

After reducing the size, the matrix $\mathbf{F}_{K\times N}$ has more rows than columns, which means that the system of Equation (3) is overdetermined. As a result, the solution $\hat{\mathbf{h}}_{N\times 1}$ can only be an approximation. The least squares method [26] is used to find an impulse response that minimizes the $\ell_2$ norm between the acquired and calculated pilots. This method does not make any assumptions about the number of paths that make up the actual channel impulse response $h(t)$ [21].

*2.2. Compressive Sensing*

The channel impulse response $h(t)$ is a sparse signal with a limited number of paths. Compressive sensing algorithms aim to minimize the approximation error of the sparse response $\mathbf{h}_{N\times 1}$ by finding the solution with a presumed number of non-zero coefficients. The FFT matrix $\mathbf{F}_{K\times N}$ is commonly used as a sensing matrix due to its good selective properties [13]. Equation (5) defines the sparse system. In the presence of noise in the received signal, the optimization problem in the noisy channel can be formulated as follows, where $\epsilon$ limits the level of noise:

$$\min \|\mathbf{h}_{N\times 1}\|_{\ell_0} \text{ subject to } \|\mathbf{F}_{K\times N}\mathbf{h}_{N\times 1} - \mathbf{H}_{K\times 1}\|_{\ell_2} \leq \epsilon, \tag{6}$$

The minimization of $\mathbf{h}_{N\times 1}$ with $\ell_0$-norm can be achieved using algorithms such as OMP [14] or CoSaMP [16]. These algorithms search for the best position of the filter

coefficients $\hat{\mathbf{h}}_{N \times 1}$ and then calculate the best approximation value using the least squares method, similar to the filter design case described earlier. If the path delays of the signal are accurately identified, the approximated impulse response will closely resemble the actual response.

Regrettably, both the OMP and CoSaMP algorithms require prior knowledge of the number of coefficients to be found. The CoSaMP algorithm aims to discover a predetermined number of paths that are not initially known. The basic implementation of OMP also seeks a fixed number of paths, although alternative stopping conditions have been proposed in the literature [18]. These conditions are based on the power of the residue in successive iterations. A reliable stopping criterion occurs when the residue power is less than the noise power; however, the receiver needs to estimate the noise level [27]. An alternative approach involves comparing the power of the residue in consecutive iterations and terminating when the value changes minimally [17,18].

This description has a flaw: the compressive detection algorithm aims to find the best approximation of the vector $\mathbf{h}_{N \times 1}$, which represents the channel response rather than the function $h(t)$ itself. Assuming that the lags between the paths are integer multiples of the sampling period and perfect synchronization between the transmitter and receiver, the number of non-zero coefficients in the discrete channel impulse response $\mathbf{h}_{N \times 1}$ corresponds to the actual number of channel paths $h(t)$ (Figure 3a). However, if these conditions are not met, the signal paths are not aligned with the grid, resulting in a higher number of non-zero coefficients in the digital impulse response $\mathbf{h}_{N \times 1}$ compared to the number of channel paths (Figure 3c). This is because the signal propagates through the limited number of tracks in the channel $h(t)$ and multiple processing paths in both the transmitter and receiver. The dashed line in Figure 3 represents the additional filtering.

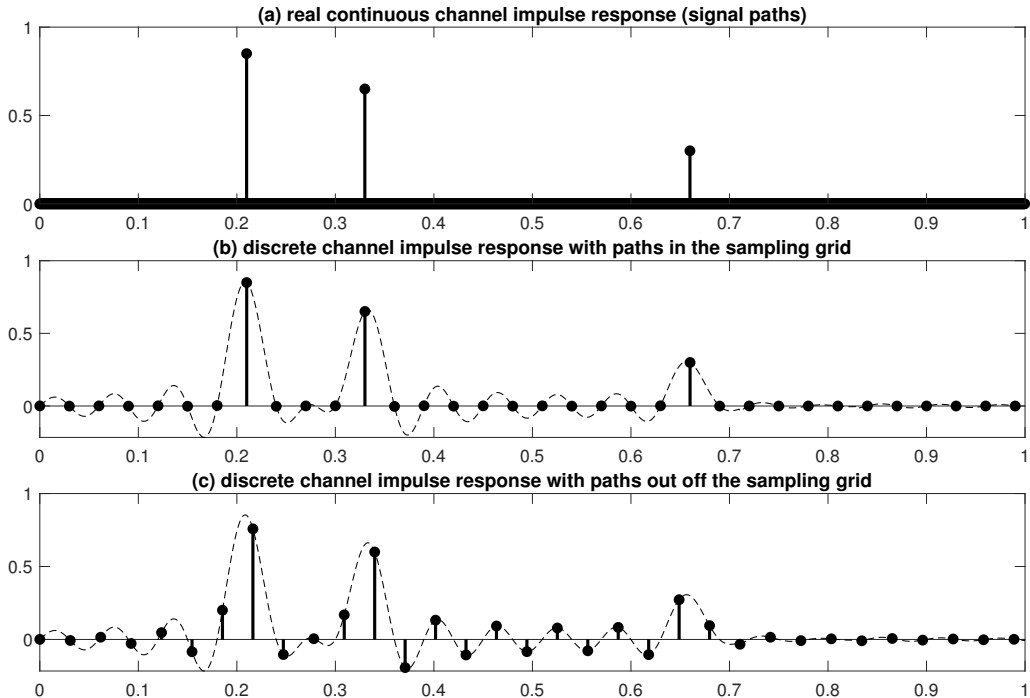

**Figure 3.** Comparison of the real impulse response of the channel and its discrete form depending on the sampling frequency.

## 3. Filtered Impulse Response of Channel

To comply with regulations and prevent interference with other systems, it is necessary to filter the signal in the transmission system into the desired frequency band. The signal processing path includes analog and digital filters. Although the impact of these filters on

equalizer design should be considered, they are partially considered in the design of the basic zero-forcing algorithm (1).

However, even if the receiver bandpass filter does not introduce distortion to the signal within the desired frequency range, it alters the signal's shape in the time domain. The channel impulse response $\mathbf{h}_{N \times 1}$ in Equation (3) is no longer sparse but remains compressible, as shown in Figure 3c. However, the compressive sensing algorithms mentioned above do not yield the expected results. The main challenge lies in accurately estimating the number of relevant channel paths [18].

One possible solution is to include band filters in the equalizer design, as described in the model presented by [20]. This approach gives rise to the following computational problem.

$$\mathbf{Y}_{K \times 1} = \mathbf{X}_{K \times 1} \odot \mathbf{F}_{K \times N} \mathbf{S}_{N \times T} \mathbf{h}_{T \times 1}. \tag{7}$$

Here, the vectors $\mathbf{X}_{K \times 1}$ and $\mathbf{Y}_{K \times 1}$ still represent the pilot signals transmitted and received in the frequency domain, respectively. As a result, the receiver does not need to perform additional measurements on the received signal. The matrix $\mathbf{F}_{K \times N}$ also remains unchanged, with the number of rows corresponding to the number of channel frequencies used in OFDM transmission and the number of columns representing the number of coefficients in the discrete impulse response of the channel.

The vector $\mathbf{h}_{T \times 1}$ represents the sparse impulse response $h(t)$ of the analog channel. However, since the system is digital, the impulse response is a discrete signal. To ensure accurate representation, the effective sampling frequency must be higher than that of the D/A and A/D converters in the transmitter and receiver. The oversampling is a conceptual operation and does not change the actual sampling frequency of the signal. In Equation (7), the matrix $\mathbf{S}_{N \times T}$ is the digital FIR filter that simulates the signal processing path of the transmission system.

At this stage, the primary advantage of the model is that the impulse response $\mathbf{h}_{T \times 1}$ in Equation (7) is sparse and can be estimated using one of the compressive sensing algorithms. Further research is focused on two areas: (1) determining the oversampling factor and the necessary density of the oversampled impulse response of the channel $\mathbf{h}_{T \times 1}$, and (2) investigating the properties of the impulse response of the FIR filter $\mathbf{S}_{N \times T}$ and the system in combination with compressive sensing algorithms—particularly the properties of the modified sensing matrix, which is now the product of two matrices.

### 3.1. Shaping Filter

According to Eldar [28], the resampling process can be divided into three main stages: interpolation, filtering, and decimation. Equation (7) implements the last two stages, which involve applying a filter to the oversampled signal $\mathbf{h}_{T \times 1}$ to obtain the band-limited impulse response of the channel and then reducing the number of samples to match the pilot signals.

The filter $\mathbf{S}_{N \times T}$ mainly represents the behavior of various components of the existing system, such as anti-aliasing filters and internal filters in sampling devices. In the field of channel modeling [29], the sinc() function is truncated to a specific length to approximate an ideal bandpass filter. Its primary purpose is to filter the signal, which in this case is the channel's impulse response, to adjust its parameters to the operating band of the transmission system. Another function that can be used in the model is the raised-cosine function [20]. We will later demonstrate that the channel estimation result is improved when the filter $\mathbf{S}_{N \times T}$ is correlated with the channel model, which can be achieved by using measured parameters of the transmission system in practice.

The signal $\mathbf{h}_{T \times 1}$ contains $F_{OS}$ times more elements compared to the channel response $\mathbf{h}_{N \times 1}$, which is obtained by transforming the frequency domain after calculating the pilots in Equation (1). The oversampling factor $F_{OS}$ can have any theoretical value, but we restrict it to powers of two in this paper.

When the oversampling factor is equal to one, oversampling has no effect, and the shaping function becomes a Kronecker delta. This function has only one non-zero element,

and the matrix $\mathbf{S}_{N \times T}$ becomes an identity matrix. Equation (7) returns to the base form given by Equation (3).

### 3.2. Sensing Matrix

Equation (6) illustrates a compressive sensing problem, in which the measured and detected signals are connected through the sensing matrix $\mathbf{A}$:

$$\mathbf{s}_y = \mathbf{A}\mathbf{s}_x + \mathbf{s}_z. \tag{8}$$

Here, $\mathbf{s}_y$ represents the measurement results, $\mathbf{s}_x$ denotes the sparse signal to be identified, and $\mathbf{s}_z$ represents the presence of noise.

If the sensing matrix satisfies the restricted isometry property (RIP), compressive sensing algorithms can provide valuable results [12,13]. If matrix $\mathbf{A}$ satisfies order $m$'s RIP, then a signal $\mathbf{s}_x$ can be reconstructed using only an $m$-sparse vector. However, computing the RIP for a given sensing matrix $\mathbf{A}$ is challenging, as it requires searching through all combinations of matrix columns. The mutual coherence parameter is often used to test the sensing matrix for sparse recovery practically [30]. For a given matrix $\mathbf{A}$, the mutual coherence is defined as the maximum absolute inner product between any two distinct columns of $\mathbf{A}$, normalized by the product of their Euclidean norms:

$$\mu(\mathbf{A}) = \mu = \max_{i \neq j} \frac{|\mathbf{a}_i^H \mathbf{a}_j|}{\|\mathbf{a}_i\|_{\ell_2} \|\mathbf{a}_j\|_{\ell_2}}, \tag{9}$$

where $\mathbf{a}_i$, $\mathbf{a}_j$ are the columns of $\mathbf{A}$. A lower mutual coherence value indicates that the columns of the sensing matrix are highly incoherent, which improves the ability of the matrix to distinguish significant values in sparse results using greedy algorithms such as OMP or CoSaMP. Although there is no strict threshold for the mutual coherence value in relation to signal sparsity $M$, the formula

$$M < \frac{1}{2}\left(\frac{1}{\mu} + 1\right), \tag{10}$$

is sometimes used to estimate the maximum number of meaningful elements in the sparse vector [12]. Once $M$ sparse elements have been identified, the predictive power of matching pursuit algorithms diminishes, and the previously identified indexes contain more information than new ones. Generally, a lower mutual coherence value indicates a better matrix for compressive sensing algorithms.

The algorithm presented in this paper utilizes a Discrete Fourier Transform (DFT) matrix, denoted as $\mathbf{F}_{K \times N}$ in Equation (5), as its base matrix. The DFT matrix performs well in algorithms that aim to find sparse signals [13,31]. When $K = N$, the columns of the DFT matrix exhibit high incoherence, resulting in a $\mu$ value of zero. This makes the matrix useful as a sensing matrix for signals with any sparsity level. However, the oversampling and filtering performed by $\mathbf{S}_{N \times T}$ in (7) affect the detection properties of the matrix. The matrix can be expressed as $\mathbf{A}_{K \times T} = \mathbf{F}_{K \times N} \mathbf{S}_{N \times T}$.

The effectiveness of the filtered sensing matrix for greedy algorithms was evaluated by examining all values in the set used to compute the maximum in Equation (9). The set elements were calculated for low-pass filters defined by the functions sinc() and srrc() (square-root-raised-cosine) with varying roll-off factors $\beta$. The most favorable outcomes were observed for the sinc() function, which is known for its narrow main lobe and discriminating properties [32]. An example of the calculation results is shown in Figure 4.

The spacing between values in Figure 4 is equal to a quarter of the sampling period, indicating an oversampling factor of $F_{OS} = 4$. The value of $\mu$ is approximately 0.9. The application of filtering negatively affects the sensing properties of the DFT matrix, making it challenging to use the filtered sensing matrix for sparse algorithms. However, high $\mu$ values are only present in adjacent columns, as shown in Figure 4. The selectivity of

the sensing matrix primarily deteriorates in neighboring elements. Despite the apparent conclusion from Equation (10), we can still utilize the provided matrix by considering the following guidelines: (1) The sparse algorithm cannot distinguish between paths that are closer than the system's sampling period, and (2) if the search algorithm detects paths that are closer than the sampling period, they may be the result of the blurring of the same path.

To clarify, the issues related to path differentiation after oversampling persist even when the receiver-side filtering is simplified to applying the sinc() function. The sinc() function is the optimal selection for performing path searching greedily, considering the effectiveness of path discrimination.

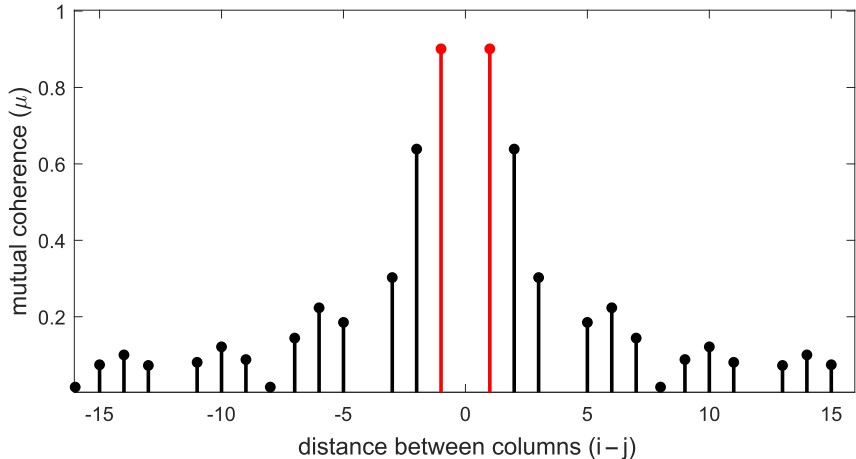

**Figure 4.** The values calculated from (9) for different distances between columns for sensing matrix **A** with 4× oversampling and filtering with sinc() function. The red color indicates the values that have the most adverse effect on the performance of the search stage of the OMP (Orthogonal Matching Pursuit) algorithm.

## 4. Estimation Errors

Equation (7) takes into account an oversampled channel impulse response $\mathbf{h}_{T \times 1}$, which accurately represents the analog signal $h(t)$ when the oversampling factor is significant enough. Using a discrete model with a high oversampling factor, we can effectively capture the delay of any propagation path in the system. Increasing the oversampling factor improves the accuracy of determining the delay of each signal path. However, this also results in increased computational complexity.

The base cause of errors in the channel estimation process in the presented method is the limited oversampling factor, which leads to inaccuracies in path delay estimation. Another source of error arises from the discrepancy between the characteristics of the low-pass filter of the system and the one used in the sensing matrix described in Equation (7).

Theoretically, it is possible to prevent both errors by ensuring that the filter used in the recovery process perfectly matches the shaping filter used in the transmission system. Additionally, the accuracy of the channel impulse response can be improved by increasing its density to the point where each path delay can be precisely determined. However, achieving the former is technically tricky, and the latter significantly increases the complexity of computations.

### 4.1. Discrete Channel

To assess the accuracy of the estimation errors, it is necessary to define two filters present in the system. The first filter, the shaping filter, represents the sampled version of the channel impulse response $h(t)$. This filter generates the channel model $\mathbf{h}_{N \times 1}$. On the other hand, the second filter is used to compute the estimated channel response $\hat{\mathbf{h}}_{N \times 1} = \mathbf{S}_{N \times T} \mathbf{h}_{T \times 1}$ in Equation (7).

Both filters simulate the discrete channel with restricted bandwidth, as proposed by the authors of [29], and can be expressed as follows:

$$g_n = \sum_{i=1}^{I} a_i \, \alpha(i, n), \quad -N_1 \le n \le N_2, \tag{11}$$

where $\alpha(i, n)$ represents the shaping function, which is sampled with a period of $T_s$:

$$\alpha(i, n) = \text{func}\left(\frac{\tau_i}{T_s} - n\right). \tag{12}$$

The parameters $a_i$ and $\tau_i$ for $i = 1, 2, ..., I$ determine the amplitude and lag of the $i$th path of the channel. The duration of the path model is limited to $N = N_1 + N_2 + 1$ samples.

Typically, the channel is represented using the sinc() function. This modeling approach is commonly implemented in Matlab® [33]. Other functions, such as the square-root-raised-cosine srrc() function, also exhibit the desired properties of limited bandwidth and decreasing amplitude in the time domain.

*4.2. Reconstruction Error*

We only consider the reconstruction error for the channel with a single path ($I = 1$) because this simplified analysis provides enough insight into the error mechanism. The Mean Squared Error (MSE) is used to measure the reconstruction error. To calculate the error, we define two functions for the replacement of func() in the model given by Equation (12): $\text{func}_1()$ generates the channel model (which describes the existing system) and $\text{func}_2()$ is used to reconstruct the channel impulse response in Equation (7) by searching for a channel response in the sparse space. If the shaping filter $\text{func}_1()$ is known, then $\text{func}_1() = \text{func}_2()$ should be used to minimize the error.

Considering the information mentioned above, the MSE can be expressed as:

$$\text{MSE} = \log_{10}\left(\frac{1}{N} \sum_{n=1}^{N} (g_n - f_n)^2\right), \tag{13}$$

and

$$g_n = \sum_{i=1}^{I} a_i \, \text{func}_1\left(\frac{\tau_i}{T_s} - n\right), \quad -N_1 \le n \le N_2, \tag{14}$$

$$f_n = \sum_{j=1}^{J} b_j \, \text{func}_2\left(\frac{v_j}{T_s} - n\right), \quad -N_1 \le n \le N_2. \tag{15}$$

The primary cause of the error is the discrepancy in the shape of $\text{func}_2()$ and $\text{func}_1()$. If the receiver cannot accurately determine the shape of the channel filter, an error occurs. The second cause of the error is the discretization of time. If the path lags $\tau_i$ and $v_j$ do not align with the grid of sampling moments (see Figure 3c), the error increases. Even if we have the same func() in both filters, but the delays differ or the difference between them is less than the sampling period $T_s$, a non-zero MSE is observed.

Let us first analyze the second source of the error. The sampling period of the signal is denoted as $T_s$. However, the sampling moments in the transmitter and receiver are shifted by $\Delta\tau = \tau_i - v_j$, which can range from 0 to $T_s/2$. If the shaping function is known and the length $N$ of $g_n$ is equal to the size of $f_n$, then there is no expected error as defined by Equation (13). The vectors of the coefficients are the same: $\mathbf{a}_{I \times 1} \equiv \mathbf{b}_{J \times 1}$ and the shift $\Delta\tau = 0$. However, when the shift is different from zero and the result of the greedy algorithm is limited to a specific number of coefficients, the error appears. For example, we present an analysis of the channel model with the sinc() function defined for $N = 128$ samples. The estimation error depends on the shift value $\Delta\tau$ and the number of estimated channel coefficients, as shown in Figure 5.

The reconstruction error values shown in Figure 5 decrease as the value of $\Delta\tau$ decreases, and more channel impulse response coefficients are determined. Increasing the oversampling ratio (corresponding to a higher density grid of values for the coefficient delays) can improve the error. However, it is essential to note that this increase in the oversampling ratio also increases the size of the sensing matrix, which in turn leads to an increase in the computational time and memory complexity of the greedy algorithm. Furthermore, we can observe that a larger number of steps of the greedy algorithm are required when a more significant number of channel impulse response coefficients are determined to decrease the MSE.

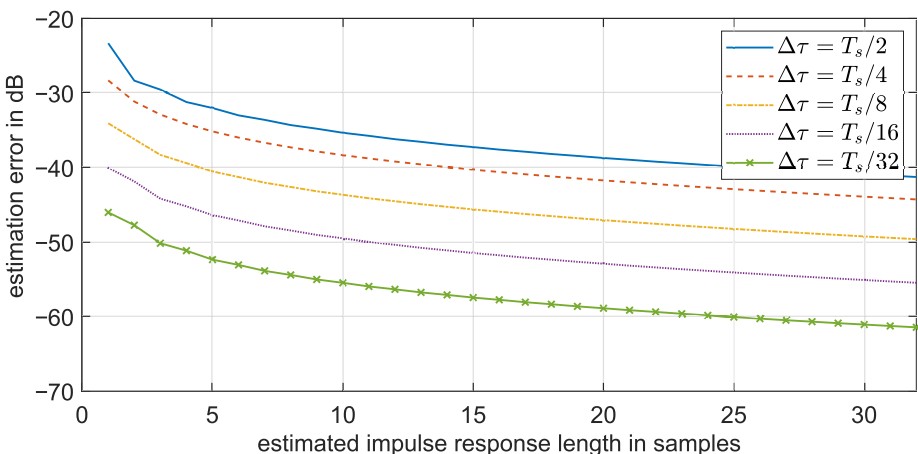

**Figure 5.** Reconstruction error when sampling moments of original and estimated channel impulse response are shifted by $\Delta\tau$.

Different interpretations can be derived from Figure 6, which illustrates the error resulting from the unmatched shaping function. Our channel modeling involved the utilization of the length-limited $\text{func}_1(x) = \sin(x)/x$ function. The error was calculated using Formula (13) and applied to the first $K$ most significant coefficients. We employed the raised-cosine function with varying roll-off factors $\beta$, as well as the function obtained through IFFT from the ideal frequency response of the channel to estimate the channel impulse response. A $\Delta\tau = T_s/2$ sampling shift was used.

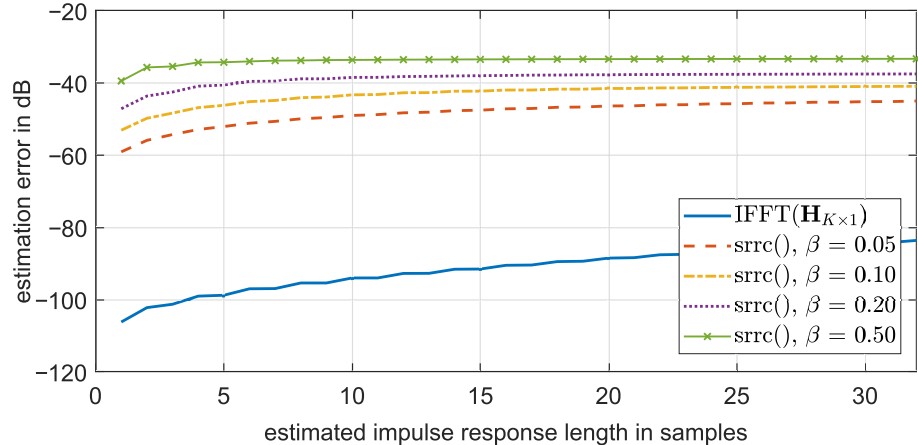

**Figure 6.** Reconstruction error caused by different shaping functions.

The error in the reconstruction procedure can be significant if the function used in Equation (7) does not match the actual function, as shown in Figure 6. This error can be comparable to the error caused by a mismatch in the sampling moments. However, the

error increases as the number of estimated coefficients increases. To minimize the error, it is advisable to detect as few channel impulse response coefficients as possible. Therefore, the greedy algorithm should terminate as soon as possible. This contradicts the earlier conclusion regarding the required impulse response length for reconstruction with matched filters.

### 4.3. Impact of Noise

This analysis does not consider the presence of unavoidable noise in the current communication systems. Although we aim to minimize the MSE in determining the channel impulse response, it is essential to note that some of the observed MSE is irreducible due to the impact of system noise. The literature, such as the work by Kang et al. (2007) [34], provides some insight into the relationship between MSE and the SNR of the channel.

The main findings of the literature analysis indicate that when using the LS (Least Squares) algorithm for channel estimation, the curve that shows the error rate (BER, Bit Error Rate or SER, Symbol Error Rate) as a function of SNR is dependent on the signal constellations implemented in the individual subchannels of the OFDM system. This characteristic is specific to the system being analyzed. However, it is possible to mitigate the noise impact by reducing the length of the estimated channel impulse response [34]. The remaining coefficients can be fixed or set to zero if the number of relevant coefficients in the time domain is limited to $L$. The resulting MSE caused by the channel SNR will decrease proportionally to the ratio of elements used $L$ to the total number of elements $N$.

Consider the OFDM system with $N$ channels. The cyclic prefix of the system has a length of $P$. We determine the unconstrained impulse response of the channel, denoted by $\mathbf{h}_{N \times 1}$, which is an $N \times 1$ matrix. This is carried out based on the training data, as shown in Equation (4). Let us assume that the error in the estimation using the ZF algorithm, denoted as $\mathrm{MSE}_0$, is solely due to noise. In that case, if we compute the impulse response again and limit its length to $P$, following the approach in [34], the error caused by noise decreases to $P/N \times \mathrm{MSE}_0$.

When employing a compressive sensing algorithm, if the estimated impulse response is constrained to a small number of coefficients $L$, which is less than the cyclic prefix length $P$, the resulting MSE caused by noise is also limited, as previously explained. In simpler terms, reducing the number of coefficients we examine leads to a decrease in the error caused by noise.

### 4.4. Summary

The preceding examination of reconstruction and estimation errors demonstrates the importance of accurately determining as many coefficients of the channel impulse response as feasible, provided that we can effectively reconstruct them.

Given the understanding that the channel response exhibits sparsity, a greedy algorithm, such as OMP, is expected to yield favorable outcomes when a few coefficients are computed. This approach helps mitigate the impact of noise. However, it is worth noting that the channel response is also compressible. In such scenarios, an algorithm is required to effectively identify a limited number of genuine paths (to minimize noise interference) while producing a significant number of reconstructed channel coefficients that closely resemble the actual ones, thus ensuring high precision.

Suppose the time response of the channel is not known, but its compressible nature is acknowledged. In that case, the OMP algorithm can be employed with a stop criterion based on the balance between the estimated MSE and noise power, as suggested by Dziwoki et al. [18]. Alternatively, if the nature of the time response is known, we recommend using the algorithm described later in this paper.

## 5. Proposed Algorithm

The above analysis focuses on determining the channel's impulse response with a resolution higher than the sampling period. In the receiver, the next step is to use a

standard one-tap equalizer for each subchannel of the OFDM system. This equalizer is created using the elements on the right side of Equation (7):

$$\mathbf{E}_{K \times 1} = \mathbf{1}_{K \times 1} \oslash (\mathbf{F}_{K \times N} \mathbf{S}_{N \times T} \mathbf{h}_{T \times 1}). \tag{16}$$

The vector $\mathbf{E}_{K \times 1}$ has the same number of elements as the number of subchannels in the OFDM system. Equation (16) consists of two operations: filtering the channel impulse response $\mathbf{h}_{T \times 1}$ with the shaping filter $\mathbf{S}_{N \times T}$ and performing the FFT operation $\mathbf{F}_{K \times N}$ on the filtered value. The filtering operation is used to create a channel model [29]. Based on the analysis results in the previous section, the best results are expected when the same shaping filter is used for data transmission and channel estimation. The filter parameters must somehow be transferred to the receiver in the existing system.

The algorithm for estimating the equalizer is derived from the analysis explained above and can be summarized as follows:

1.  Create the sensing matrix $\mathbf{A}_{K \times T} = \mathbf{F}_{K \times N} \mathbf{S}_{N \times T}$ as in (7).
2.  Use the OMP algorithm [14] to estimate channel impulse response $\mathbf{h}_{T \times 1}$ from (7). Extend the search step with neighbors as described below in the first modification.
3.  Calculate the equalizer coefficients using Formula (16). Use a shaping filter according to the second modification described below.

The significant modifications that have been introduced enhance the determination of the equalizer coefficients in the transmission system when operating in a noisy environment.

The first issue that the implementation addresses is the problem of the oversampling factor. In a noise-free system, a higher factor value yields better results. However, this also increases computational complexity due to the larger matrix $\mathbf{A}$ size. Taking noise into account, it is essential to note that the oversampled detection matrix $\mathbf{A}$ has poor distinguishing properties (see Figure 4), and the estimation error is low even with a high resolution of the oversampled signal (refer to Figure 5). Going beyond a value of 8 of an oversampling factor does not significantly enhance the quality of the search stage, but it comes at a higher computational complexity cost.

Based on the findings presented earlier in Figure 5, it is anticipated that the accuracy of determining the delay values will be compromised by limiting the oversampling factor. However, we assumed that the actual maximum value obtained during the search phase of the OMP algorithm is close to the values determined. This assumption leads to a significant modification of the algorithm: The matrix utilized in the LS algorithm stage of the OMP is composed of the maximum value found in each iteration of the search phase, along with the adjacent columns of the sensing matrix. A similar approach is proposed for the OMP block algorithm [35], which assumes that the result is sparsely grouped.

The algorithm undergoes a second significant modification in which the sensing matrix used in the OMP algorithm differs from the matrix used to calculate the equalizer coefficients. These matrices have varying shaping filters, denoted as $\mathbf{S}_{N \times T}$. The shaping filter used to calculate the equalizer in step 3 aims to closely match the shape of the transmission path over time, as observed in Figure 6. On the other hand, the OMP algorithm in the search phase utilizes the sinc() function as a shaping filter, using a rectangular window during the oversampling process. Though this provides the best possible path discrimination properties in an upsampled signal, it still needs improvement, as demonstrated in Figure 4.

## 6. Numerical Simulation

### 6.1. Simulation Parameters

We evaluated the algorithm in a simulated transmission system environment. We employed OFDM modulation with the QPSK (Quadrature Phase Shift Keying) constellation on each of the 128 channels. The size of the cyclic prefix was set at 32 samples, which was considered adequate based on the measurements we carried out using physical devices in different working environments, including indoors and outdoors, in urban and outdoor areas.

Most simulations were conducted on a channel with two paths with similar attenuation levels. This particular channel was the most challenging environment during the measurements. The distance between the two paths was set to be within the 0.5 to 2.5 sampling periods, and the sampling moment varied within one sampling period. This channel type is relatively straightforward in the time domain but presents significant difficulties in equalization in the frequency domain due to deep fades. In the simulation model, the phase of the second path was adjusted from 0 to $2\pi$ depending on the specific realization of the simulation. As a result, the deep fading was shifted across the entire frequency band of the transmission system.

We conducted simulations on 100 channels consisting of two paths to obtain statistically significant results. The parameters used for these channels were as described earlier. For each channel, we transmitted a total about 100,000 symbols, which included one frame with pilots and 800 frames with random data. The channel model for each case was created based on the literature [33], where the paths were filtered using the sinc() function.

We compared the channel estimation obtained using the oversampling algorithm presented here with the standard OMP algorithm. We also compared the results obtained using the simple zero-forcing equalizer with noise estimation (ZF MMSE in the figures) and the data obtained when the receiver knows the simulated channel state information (known as CSI in the figures). The known CSI result represents the performance limit the equalizer aims to achieve based on the estimated channel.

We modified the parameters for particular simulations. Initially, we examined the channel model with data paths in the sampling grid. We emphasized the importance of testing actual channels with values, not on the grid, and considering the possibility of unnecessary modifications to the proposed algorithm. Subsequently, we investigated our analysis using a channel model that employed a different function from the standard sinc(). It was demonstrated that the function utilized for equalizer calculations should closely resemble the channel model. Therefore, we conducted simulations using a modified channel modeling algorithm. Finally, we evaluated the performance of the algorithm presented in a different environment from our measurements by employing COST-207 channel models [21].

### 6.2. Simulation Results

Figures 5–8 show the results of the simulation. Each data point in the figures represents the average of 100 distinct channel implementations, each with the specified structure for the corresponding set of simulations.

The initial results depicted in Figure 7 entail a comparison between the basic OMP algorithm and the algorithm that incorporates oversampling (referred to as $F_{OS}$ = X in the Figure, where X represents the oversampling factor) for channels with two paths and a delay of one or two sampling periods between paths. These channels fall within the previously defined range, but the delays always conform to the sampling grid. As observed in the figure, the basic OMP algorithm performs exceptionally well, and while oversampling does not yield any improvement, it also does not lead to significant degradation. All versions of the OMP algorithm aim to identify the two paths (which serves as a stopping criterion). Upon closer examination of the results, it is evident that the discrepancy between the estimated and known channel response is approximately 0.3 dB at an SNR of 25 dB.

The results shown in Figure 8 are obtained from a more realistic model where the paths are not located on the sampling grid. The standard OMP algorithm performs poorly with this channel type when searching for only two paths. To address this issue, we have implemented an additional stop criterion in the OMP search algorithm, which analyzes the power of the residual error. As a result, the number of coefficients in the channel's impulse response varies between simulations. The outcome (OMP stop) is significantly better than the basic OMP, but it is still not satisfactory, as per our previous work [18].

The performance of the oversampling algorithm is satisfactory in this model. The estimated channel shows a difference of approximately 0.7 dB at an SNR of 25 dB compared

to the known CSI result, and the difference becomes even smaller for low SNR values. The analysis presented in Figure 5 confirms that the comparison between standard and oversampled OMP algorithms is consistent. After oversampling, all channels reconstructed from the identified paths were filtered using the shaping filter that matches the channel model. However, the sensing matrix has poor discriminating properties, resulting in a negligible gain for higher oversampling factors.

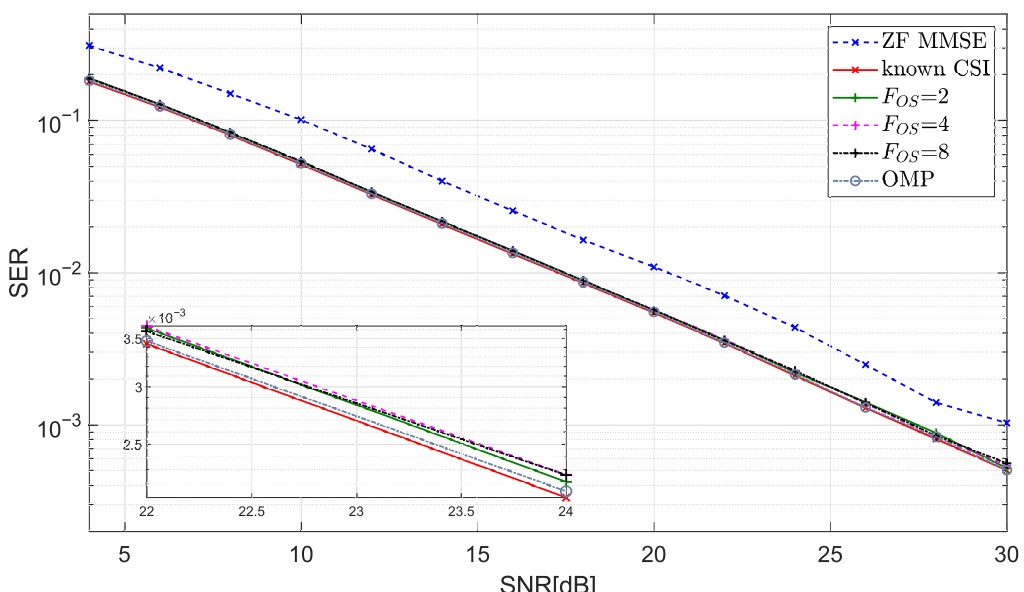

**Figure 7.** Comparison of tested algorithms on channels with delays in the sampling grid (channel with 2 paths, results averaged over 100 channel realizations).

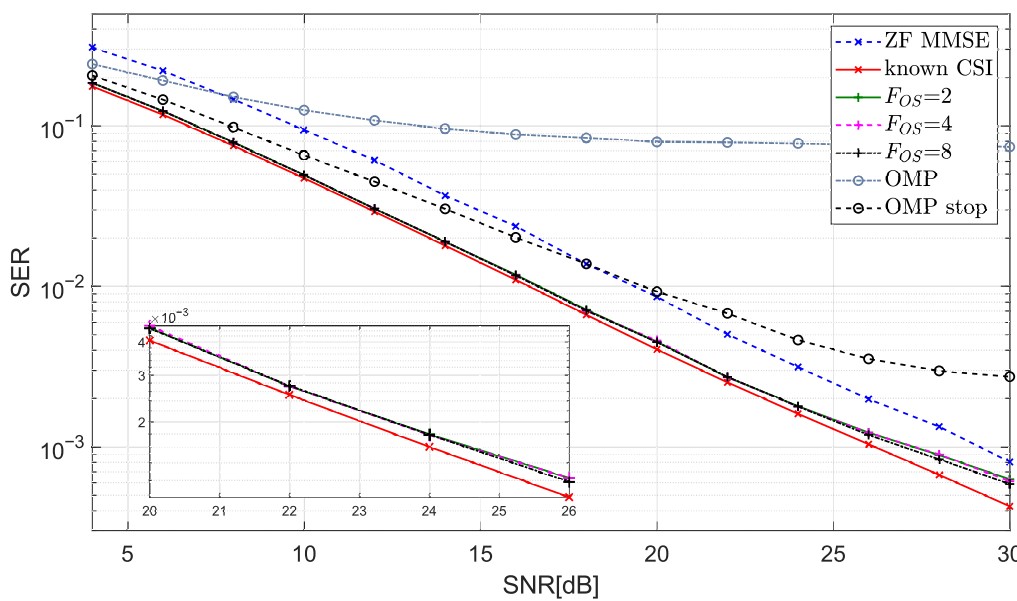

**Figure 8.** Comparison of tested algorithms on channels with delays out of the sampling grid (channel with 2 paths, results averaged over 100 channel realizations).

The results shown in Figure 9 demonstrate the impact of mismatch between the shaping filter in the transmission system and the equalizer. These results align with the analysis presented in Figure 6. When the signal-to-noise ratio (SNR) is low, the equalizer filter can differ from the one used in the transmission system, resulting in similar symbol error rates (SERs) for all shaping filters. However, the matching between the two filters becomes necessary

as the SNR increases. In this case, oversampled models with a factor of four were used (see Figure 8). The channel was generated using the sinc() function, while the equalizer utilized sinc() and srrc() filters with different values of $\beta$. As expected, the results indicate that the closer the shaping filter is to the channel model, the fewer errors are detected.

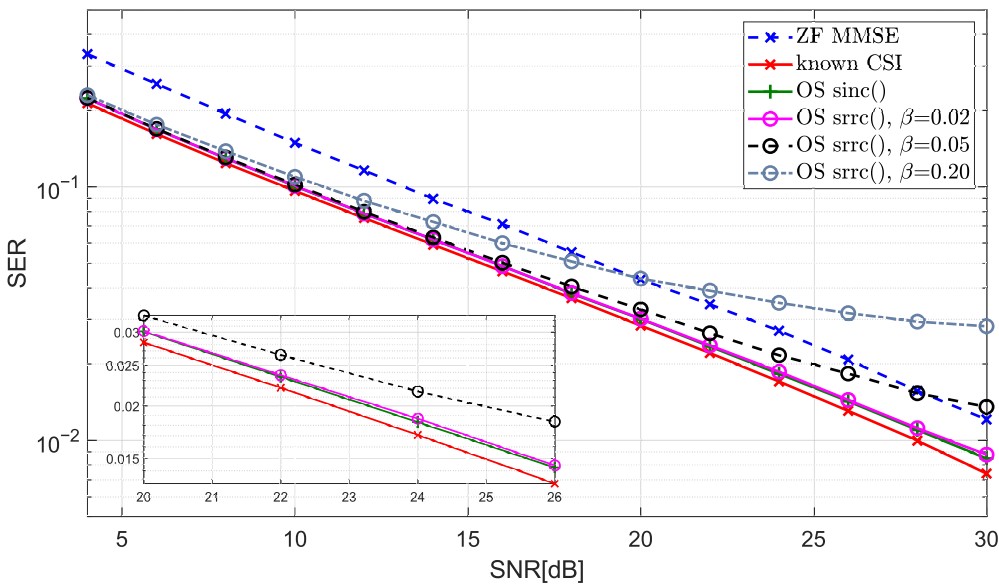

**Figure 9.** Comparison of tested algorithms on channels with delays out of the sampling grid and unmatched shaping filter—channel modeled with the sinc() function (channel with 2 paths, results averaged over 100 channel realizations).

The last result involved a modification in channel type. Specifically, the COST-207 model [21] was used to generate 30 channels, each consisting of 6 paths. Gain values from the COST-207 model were adopted, but absolute delay values were adjusted to fit within the range of the cyclic prefix in the simulation environment. It is important to note that the relative relationships between the delay values were maintained. Additionally, the number of paths to be explored in the OMP algorithm was increased to six.

The results for the COST-207 model shown in Figure 10 are comparable to those obtained for the channel with only two paths. Notable differences can be observed for higher SNR values. In this case, the algorithm aims to identify six significantly noise-impacted paths. The standard OMP algorithm once again fails to perform adequately (path delays deviate from the grid), emphasizing the importance of selecting an appropriate shaping filter (the channel was simulated using the sinc() function).

The disparities in the performance of the evaluated channel equalization algorithms become more noticeable as higher SNR levels are applied. In this range, a more accurate channel correction by the equalizer will enable efficient transmission, even with correction codes that have reduced redundancy [36].

### 6.3. Computational Complexity

The computational complexity of the presented algorithm primarily stems from the OMP algorithm itself. An analysis of OMP's complexity can be found in the literature [37]. In brief, it depends on three key factors: the size of the sensing matrix (search stage), the number of paths searched (matrix inversion in the LS algorithm stage), and the number of iterations.

The search stage is the most time-consuming. Oversampling increases the number of computations in the search stage in proportion to the oversampling factor $F_{OS}$. Furthermore, the use of adjacent coefficients increases the number of elements calculated in each iteration of the LS algorithm by a factor of three. The number of iterations remains unchanged compared to the baseline OMP.

Thus, while achieving greater accuracy in determining channel characteristics, the trade-off is increased numerical complexity. Interestingly, it turns out that it is possible to significantly reduce the computational effort by selectively oversampling specific parts of the sensing matrix, as shown in [38].

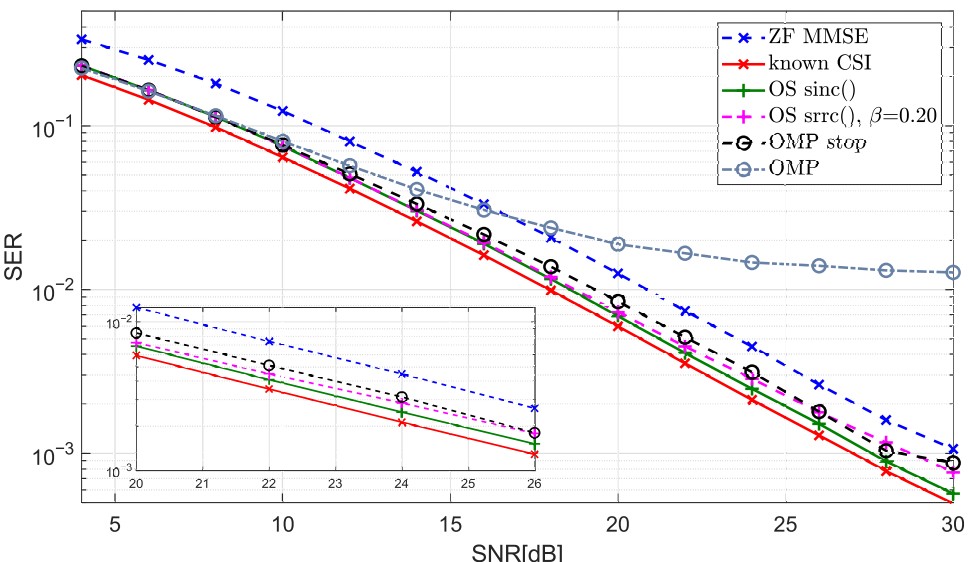

**Figure 10.** Comparison of tested algorithms on example COST-207 channel (channel with 6 paths, results averaged over 100 channel realizations).

## 7. Conclusions

The paper concentrates on channel equalization based on its physical nature. The work started on OMP algorithm implementation to the recognized channel model. However, the OMP algorithm is effective only when the delays of the channel paths align with the sampling devices' grid. It is possible to increase the number of elements searched by the OMP. However, it takes the algorithm away from the channel's physical model, and in a noisy environment, enhances the effect of noise on the result.

Another approach presented in this paper is to search for channel coefficients, or more precisely, paths describing delays, with a resolution higher than the sampling grid distance. The resolution of the model is increased with an oversampling algorithm using interpolation and filtering. Then the OMP algorithm, again with only a few elements to find, can be implemented on the oversampled channel model.

Creating the oversampled model is quite simple, but a deeper analysis presented here showed that the type of filter used in the algorithm is essential. This shaping filter should match the characteristics of the signal transmission path for the best results. Theoretically, it is possible to increase the channel resolution indefinitely, but it reduces discrimination properties of the sensing matrix and does not improve the result of channel reconstruction.

We proposed modifications to the OMP algorithm based on the analysis to achieve better channel estimation in a noisy environment. Primarily, we distinguished filters used in the different stages of computation. We used various ones during the search stage of the OMP and during channel reconstruction before the equalizer coefficients' calculation. The second modification, which treats the results of the search stage as elements of groups, also increased the algorithm's overall performance.

We have focused on the best possible channel reconstruction, especially in our transmission environment with two relative paths, but the algorithm was also successfully tested with more complex channels. The problem of computation complexity was put in the background during the research. Analyzing the proposed modifications, we can say that their impact on the algorithm already working on oversampled signals is negligible. However, the oversampling itself increases the number of computations because of the increased size

of the sensing matrix. The successive delay approximation algorithm presented in [38] will help save computing resources.

Some questions still need to be addressed in our analysis and additional research. Further mathematical analysis would be helpful to define the best oversampling factor precisely and the required quality of the shaping filter matching the channel characteristics. Additionally, analysis is needed to determine the shaping filter characteristics for a given transmission system.

**Author Contributions:** Conceptualization, G.D.; Methodology, M.K. and G.D.; Software, M.K. and J.I.; Validation, W.S., A.D., W.F., W.I., P.K., P.Z., P.S. and M.R.; Formal analysis, J.I. and W.S.; Investigation, M.K.; Resources, P.S.; Writing—original draft, M.K.; Writing—review & editing, G.D., J.I. and W.S. All authors have read and agreed to the published version of the manuscript.

**Funding:** This work was supported by the Ministry of Science and Higher Education funding for statutory activities (BK-238/RAu13/2023).

**Data Availability Statement:** Data are contained within the article.

**Conflicts of Interest:** The authors declare no conflicts of interest.

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
