# Peer review of "Equalizer Parameters’ Adjustment Based on an Oversampled Channel Model for OFDM Modulation Systems"

_electronics, doi:10.3390/electronics13050843_

Round 1

Reviewer 1 Report

Comments and Suggestions for Authors This work is interesting. Methods for equalizer filter parameters choice of an oversampled channel model for low transmission errors are suggested for OFDM modulation systems. A specific upsampling filter and a specific downsampling filter are used to searching the path delay and channel impulse response, and then the path delay and nearest neighbors are used in the compressive sensing algorithm. I have some concerns: Major issue: Photos of the experimental system may be given. Minor issue: In the abstract, the details of the modifications to the compressible channel’s impulse response reconstruction algorithm may be summarized. The computation cost of the algorithms may be given. Overall suggestion: Major revision

Reviewer 2 Report

Comments and Suggestions for Authors

This work is very interesting and useful for the engineering community. The work as presented is scientifically excellent. However, adding some information may help to improve the quality of the manuscript.
1. Please highlight the advantage and disadvantage of using the approach of parameter adjustment presented in this work while comparing to the other ones.
2. Is there specific limitation of the proposed approach on the order of OFDM modulation signal? Please specify if there’s any.
3. For high operating frequencies, the equalizing work of modulated signal becomes critical. What’s your thought on adopting the proposed method for high-frequency systems?

Comments on the Quality of English Language

Some work needs to be done to correct the grammar and optimize the presentation of the data. Please go through the manuscript again to shorten some long sentences. Also, please check the consistency of the figures.

Round 2

Reviewer 1 Report

Comments and Suggestions for Authors

All my concerns are resolved, no further comments.